# Vertical Hot-Melt Extrusion: The Next Challenge in Innovation

**DOI:** 10.3390/pharmaceutics17070939

**Published:** 2025-07-21

**Authors:** Maël Gallas, Ghouti Medjahdi, Pascal Boulet, Victoire de Margerie

**Affiliations:** 1Rondol Industrie, 2 Allée André Guinier, 54000 Nancy Cedex, France; mael@rondol.com; 2Institut Jean Lamour (IJL), Université de Lorraine, CNRS, 2 Allée André Guinier, 54000 Nancy, France; ghouti.medjahdi@univ-lorraine.fr (G.M.); p.boulet@univ-lorraine.fr (P.B.)

**Keywords:** vertical hot-melt extrusion, drug delivery systems, extrusion processing parameters, amorphous solid dispersions, solubility, bioavailability

## Abstract

**Background/Objectives:** Hot-melt extrusion (HME) has become a key technology in pharmaceutical formulation, particularly for enhancing the solubility of poorly soluble Active Pharmaceutical Ingredients (APIs). While horizontal HME is widely adopted, vertical HME remains underexplored despite its potential benefits in footprint reduction, feeding efficiency, temperature control, and integration into continuous manufacturing. This study investigates vertical HME as an innovative approach in order to optimize drug polymer interactions and generate stable amorphous dispersions with controlled release behavior. **Methods:** Extrusion trials were conducted using a vertical hot-melt extruder developed by Rondol Industrie (Nancy, France). Acetylsalicylic acid (ASA) supplied by Seqens (Écully, France) was used as a model API and processed with Soluplus^®^ and Kollidon^®^ 12 PF (BASF, Ludwigshafen, Germany). Various process parameters (temperature, screw speed, screw profile) were explored. The extrudates were characterized by powder X-ray diffraction (PXRD) and small-angle X-ray scattering (SAXS) to evaluate crystallinity and microstructure. In vitro dissolution tests were performed under sink conditions using USP Apparatus II to assess drug release profiles. **Results:** Vertical HME enabled the formation of homogeneous amorphous solid dispersions. PXRD confirmed the absence of residual crystallinity, and SAXS revealed nanostructural changes in the polymer matrix influenced by drug loading and thermal input. In vitro dissolution demonstrated enhanced drug release rates compared to crystalline ASA, with good reproducibility. **Conclusions:** Vertical HME provides a compact, cleanable, and modular platform that supports the development of stable amorphous dispersions with controlled release. It represents a robust and versatile solution for pharmaceutical innovation, with strong potential for cost-efficient continuous manufacturing and industrial-scale adoption.

## 1. Introduction

Hot-melt extrusion (HME) technology is well known in the food industry (for example, in pasta and chocolate) and is also widely used for producing/shaping many materials (plastics, aluminum, composites), but it only saw application in FDA approval for drug production in the early 2000s [1].

HME is a process which involves mixing drugs and carrier excipients in a metal tube (the barrel), where they are processed at a specific speed, temperature, pressure, and screw configuration. The mix is then pushed through a die (which gives the final shape) and cooled down (through air or water cooling), readying the product for further downstream processing [2,3].

In recent years, hot-melt extrusion (HME) has received widespread interest as a continuous manufacturing technology for producing various drug delivery systems including pellets, tablets, films, implants, and nano-delivery systems [4,5,6,7,8].

The main advantages of HME are that it is a solvent-free, high-quality, and cost-effective process [9,10].

When it comes to enhancing the solubility and bioavailability of a given API, the selection of excipients as well as the optimization of key extrusion process parameters is of paramount importance. This is especially key to lowering the shear forces needed or the temperatures required to properly mix the final product, while avoiding thermal degradation. In addition, controlling these process parameters to exhibit the smallest possible deviation is crucial for increasing the effectiveness of the interference of the API with the functionality of the other components in the formulation [9,10,11,12,13].

Varying parameters can influence the uniformity of mixing, the physical state of the incorporated drug, and/or the drug release profile from the drug delivery system [14,15].

HME in a horizontal format has already demonstrated its capability in offering specific advantages to address challenging pharmaceutical applications (more efficient for the patients, more easily compliant with FDA requirements such as CFR 21, and less capital-intensive for the industry). Thanks to this demonstrated effectiveness, over 20 marketed drug-containing pharmaceutical products have been successfully manufactured while using HME, most of which are amorphous solid dispersions (ASDs) that comprise blends of drugs and polymers and that have demonstrated improved bioavailability in vivo as compared to crystalline systems [16,17,18,19].

For example, Kaletra, a combination of lopinavir and ritonavir, was developed by Abbott and registered in the USA in 2000 for the treatment of HIV infections. The original soft gels were subsequently replaced with coated tablets for Kaletra during the HIV epidemic in Africa in 2006. These tablets were manufactured using the original Rondol HME horizontal extruder, which merges the lopinavir/ritonavir with co-polyvinyl ketone and extrudes it into granules that are further processed into tablets. Thanks to this innovative process, the tablets could be stored at room temperature and the daily dose was reduced to four tablets [20,21].

Since then, several notable HME-based formulations have been developed for the treatment of cardiovascular and infectious diseases—and they involved intense collaboration between academic and industrial partners [19,22].

However, the extrusion process in its original horizontal format could still be optimized with the aim to further minimize the quantities of APIs, improve the versatility of the extrusion process, and make the operator interface, the cleaning process, and the GMP compliance more user-friendly.

This is how the idea emerged to change the preferred process flow direction from the horizontal position to top-down in the vertical axis with a specific focus on pharmaceutical applications [23].

The developed system consists of a 10.5 mm diameter co-rotating twin-screw extruder with a 40:1 length-to-diameter (L:D) ratio. Its compact footprint supports process integration in small GMP suites. Modular universal feed ports accommodate powders, granules, liquids, gases, or volatile additives at multiple barrel locations—both vertically and laterally—via plug-in interfaces. This allows for the flexible incorporation of APIs with diverse physicochemical properties and excipients such as plasticizers, flavors, or functional additives, and supports precise material handling without hardware reconfiguration.

The vertical layout also enhances cleanability and hygienic design. Material accumulation in dead zones—common in horizontal systems—is minimized. All internal geometries are designed to prevent entrapment, and removable stainless-steel barrel liners simplify disassembly, cleaning, and validation. The vertical orientation facilitates the gravity-assisted evacuation of residual material, reducing cleaning time and operator variability [16].

Moreover, vertical extrusion improves process stability and flow control. APIs and polymer carriers often exhibit cohesive or temperature-sensitive properties. Horizontal extruders, lacking gravity assistance, rely solely on mechanical conveying, which can lead to inconsistent feed rates, residence time fluctuations, or material retention during shutdown. By aligning flow with gravity, the vertical system enables more uniform throughput and minimizes post-extrusion waste.

The system also supports various downstream configurations. Depending on the targeted dosage form, it can be equipped with strand, sheet, rod, filament, or film-casting dies. Post-die processing units such as cooling belts, pelletizers, film winders, calendaring devices, or implant cutters are fully compatible with the vertical setup, enabling seamless, in-line shaping and solidification. This configuration ensures precise control over the post-extrusion cooling and shaping stages, which is especially critical for formulations transitioning sharply between molten and solid states.

For example, in the previous work, hydroxychloroquine was reformulated while using BASF Soluplus as a model polymeric carrier with both horizontal and vertical 10.5 mm twin screw extruders [19]. And the demonstration was made of an improved robustness with the vertical extruder in comparison to the horizontal one. The reduced variation in process parameters with the vertical extruder will therefore enable high-performance continuous manufacturing with the minimum waste of (expensive) raw materials.

Further validation was conducted with Lumefantrine, the reference API in the first-line therapy used today for malaria in children—Coartem, which is a taste-masked dispersible tablet formulation co-developed by the Medicines for Malaria Venture and Novartis. The vertical process enabled improved solubility and stability under challenging environmental conditions, with a potential reduction in dose strength and administration frequency [24].

Also, in the case of acetylsalicylic acid (ASA), vertical extrusion enabled the formation of amorphous solid dispersions with 45% *w*/*w* API in Soluplus^®^. The amorphous state was maintained for at least six weeks under ambient conditions, suggesting improved physical stability for this thermosensitive model drug [25].

Several ongoing proof-of-concept (POC) studies are utilizing the vertical extrusion platform to develop pediatric medicines and formulations for rare diseases. These initiatives focus on enabling low-dose, taste-masked, or controlled-release dosage forms produced in small, GMP-compliant batches. Advanced analytical techniques such as small-angle X-ray scattering (SAXS) and Pair Distribution Function (PDF) analysis are also being employed to investigate API–polymer interactions at the nanoscale, providing structural insights into formulation behavior and performance.

## 2. Materials and Methods

The excipients used for the experimental trials included

Soluplus^®^ (BASF, Ludwigshafen, Germany)as a primary polymeric carrier;Kollidon^®^ 12 PF (BASF, Ludwigshafen, Germany)as a plasticizer and binder.

Two grades of acetylsalicylic acid (ASA), both supplied by Seqens (Ecully, France), were used to evaluate the influence of API properties on extrusion behavior and dispersion performance. Formulations were prepared with increasing API loads of 20%, 30%, and 50% *w*/*w* in Soluplus^®^, in order to assess the impact on processability and physical stability.

Formulations were premixed for 15 min in a Turbula^®^ WAB rotary mixer (WAB, Muttenz, Switzerland) to ensure homogeneity prior to extrusion, then fed through a Rondol vertical 10.5 mm twin-screw extruder with a 40:1 L/D ratio. The screw configuration included standard conveying elements as well as two kneading zones located in barrel zones 3 and 5, each combining alternating 60°–90°–60° offset kneading blocks with distributive mixing elements to ensure uniform API dispersion.

The barrel was divided into eight independently controlled heating/cooling zones. For all trials, the feed zone and zone 1 were maintained at 40 °C to ensure stable feeding conditions. Zone 2 was set at 90 °C. Zones 3 to 7 were maintained at a uniform temperature of either 115 °C or 120 °C depending on the formulation tested. The final zone (zone 8) was set at 100 °C to enable controlled pre-discharge cooling. The die was independently heated and maintained at 100 °C for all trials to ensure smooth material flow and controlled shaping at the point of discharge.

Material feed was achieved through a volumetric feeder mounted on the top port. No liquid addition was used during the trials. For each run, the system was flushed with a placebo formulation prior to API introduction. The screw speed was set at 100 rpm, and the feeding screw operated at 30 rpm, corresponding to a total feed rate of approximately 300 g/h. All trials were performed under ambient laboratory conditions.

Extrudates were collected as strands, cooled using a fixed-temperature cooling block, and subsequently cut using an in-line pelletizer.

Samples were then characterized using powder X-ray diffraction (PXRD) on a D8 Advance diffractometer (Bruker, Karlsruhe, Germany) to evaluate crystallinity and solid-state transitions of the extrudates. Additional dissolution tests were conducted according to the USP II paddle method in simulated gastric fluid (pH 1.2) and simulated intestinal fluid (pH 6.8), with sampling intervals up to 120 min. HPLC was used to quantify API release.

To understand more precisely what takes place at the microstructural level during the extrusion process, we carried out small-angle X-ray scattering (SAXS) experiments on these samples. The measurements were performed on a Xeuss 2.0 system (Xenocs, Grenoble, France). The analysis focused on determining the radius of gyration of dispersed domains, based on established SAXS theoretical frameworks and data interpretation methods developed by Porod and Guinier [26,27]. Data were collected in air, with powder samples encapsulated between two Kapton foils. The scattering contribution from the Kapton alone was subtracted from all spectra. Measurements were performed on the raw materials—pure acetylsalicylic acid, Soluplus, and Kollidon 12 PF—and on two extruded samples processed at 115 °C: one containing 30% ASA and 70% Soluplus, and the other 30% ASA, 65% Soluplus, and 5% Kollidon.

## 3. Results and Discussion

### 3.1. Process Observations and Extrudability

All ASA/Soluplus-based formulations were processed successfully under the selected conditions using the vertical 10.5 mm twin-screw extruder. The extrusion process remained stable across a range of drug loadings, with strand formation observed to be continuous and consistent at 20%, 30%, and 50% *w*/*w* API concentrations. No significant variation in torque or backpressure was detected as ASA content increased, and the screw configuration was maintained without modification throughout the trials. No bridging or feeding blockages occurred during any experiment, indicating efficient material conveyance and feeder compatibility. The extruded strands exhibited uniform surface morphology, with no visual evidence of phase separation, discoloration, or surface tackiness.

These results show that the vertical configuration provided a consistent strand formation even at high drug loadings without requiring major process adjustments nor the addition of processing aids such as plasticizers or flow enhancers, and it also facilitated stable material flow and simplified product discharge. All this suggests a robust and reproducible process environment for thermosensitive and cohesive pharmaceutical formulations.

### 3.2. Solid-State Structure and Stability 

Solid-state characterization was conducted using powder X-ray diffraction (PXRD) to assess the crystallinity of the extruded formulations. Although the PXRD diffractograms are not shown here, the data have been previously published in SPE Polymers [25]. In that study, formulations containing 20%, 30%, and 50% *w*/*w* of acetylsalicylic acid (ASA) in Soluplus^®^, processed at 120 °C, were found to be fully amorphous at the initial time point (T_0_). After three months of storage under ICH-accelerated conditions (40 °C/75% RH), the 20% and 30% formulations retained their amorphous characteristic, while the 50% formulation showed only slight recrystallization.

These results emphasize the stabilizing capacity of the polymer matrix and the influence of drug loading on long-term physical stability. Even at 50% drug content, the formulation remained predominantly amorphous, demonstrating the potential of the vertical extrusion platform to generate stable amorphous solid dispersions across a wide range of API loadings—provided that formulation parameters remain within appropriate limits.

### 3.3. Dissolution Behavior

In vitro dissolution studies were conducted at 37 °C in 0.1 N HCl to assess the release profile of ASA from the extruded formulations. As shown in Figure 1, formulations processed at 120 °C exhibited a clear correlation between drug loading and release kinetics. Lower API concentrations (20% *w*/*w*) led to a faster and more complete ASA release, while higher loadings (50 % *w*/*w*) were associated with slower dissolution rates.

These results suggest that reducing the amount of drug in the formulation does not necessarily compromise therapeutic availability and may, in fact, enhance release efficiency. The presence of Soluplus^®^ in the matrix appears to contribute to a more controlled release profile, likely due to its amphiphilic characteristic and ability to inhibit drug crystallization. Such behavior is particularly advantageous for chronic low-dose ASA applications, including those in cardiovascular and cancer prevention therapies, where sustained release and reduced variability are desirable.

A comparison between batches “2020” and “2080” further demonstrated batch-to-batch consistency and the robustness of the extrusion process. Batch “2080” consistently outperformed “2020” across all drug concentrations, achieving higher cumulative release percentages, particularly at lower API loads.

These findings indicate that vertical extrusion can enable the development of controlled-release systems through the modulation of drug loading. In the context of chronic therapies such as low-dose ASA for cardiovascular protection or cancer prevention, the ability to fine-tune release profiles while reducing daily doses may provide clinical benefits in terms of adherence and side-effect management.

### 3.4. SAXS Characterization

Small-angle X-ray scattering (SAXS) analysis was performed to investigate the internal microstructure of both raw materials and extruded formulations. As illustrated in Figure 2, Porod analysis at high q values revealed that pure ASA, Soluplus^®^, and Kollidon^®^ 12 PF displayed a q^−4^ slope in the log–log plot of intensity versus scattering vector, consistent with smooth particle surfaces and minimal internal porosity.

The extruded formulations exhibited a similar global Porod slope, but with shorter linear regions and slight inflections near q ≈ 10^−1.7^, suggesting the emergence of structural heterogeneity or incipient porosity induced by processing. These observations indicate that while the bulk morphology remains smooth, hot-melt extrusion introduces nanoscale changes in the internal organization of the matrix.

To further assess structural rearrangement, Guinier plots (Figure 3) were used to estimate the radius of gyration (Rg) of each sample.

As shown in Table 1, the extruded formulations had significantly lower Rg values (30–41 Å) compared to the raw materials (≈60–62 Å). This reduction reflects nanoscale densification and improved homogeneity of the drug–polymer system following extrusion. The ternary formulation containing 5% Kollidon^®^ 12 PF exhibited a slightly higher Rg than the binary ASA/Soluplus^®^ formulation, indicating a modified internal structure due to the presence of the second polymer.

These SAXS results suggest that vertical hot-melt extrusion may promote nanoscale densification and improved molecular dispersion within the polymer matrix. Such structural modifications could potentially contribute to enhanced physical stability and modified dissolution behavior, although further investigation would be needed to confirm these effects.

Overall, this work demonstrates that vertical extrusion enables the manufacture of high-quality amorphous solid dispersions of ASA with consistent processing and tunable release characteristics. These findings position vertical HME as a robust and adaptable platform for developing polymer-based formulations of moderately heat-sensitive APIs, with potential application in continuous manufacturing, pediatric dosage forms, and orphan drugs, particularly where small-scale GMP-compliant production is required.

## 4. Conclusions

Vertical HME represents a major advancement in pharmaceutical formulation, offering both practical and technical advantages such as its compact design, gravity-assisted product flow, precise modular feeding, optimum process control, and enhanced cleanability.

Vertical HME can also slightly reduce the particle size of the API within the extruded product. This reduction may enhance drug surface area and dissolution behavior, further contributing to improved bioavailability.

Such improvements are expected to support the development of stable amorphous dispersions with controlled release profiles. Moreover, this strategy may offer a route to reducing both capital and operational costs, while preserving therapeutic efficacy and limiting secondary effects.

Altogether, vertical HME offers a robust and versatile tool for pharmaceutical innovation and makes it a promising platform for both exploratory research and industrial-scale applications with the opportunity to standardize HME across a broader range of drug products.

## Figures and Tables

**Figure 1 pharmaceutics-17-00939-f001:**
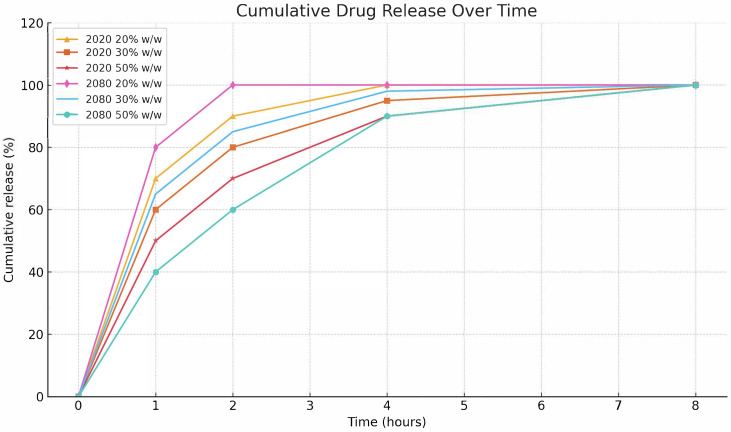
Cumulative dissolution profiles of extruded Aspirin formulations (120 °C) in 0.1 N HCl at 37 °C. Comparison between batches “2020” and “2080” at 20%, 30%, and 50% *w*/*w* ASA in Soluplus.

**Figure 2 pharmaceutics-17-00939-f002:**
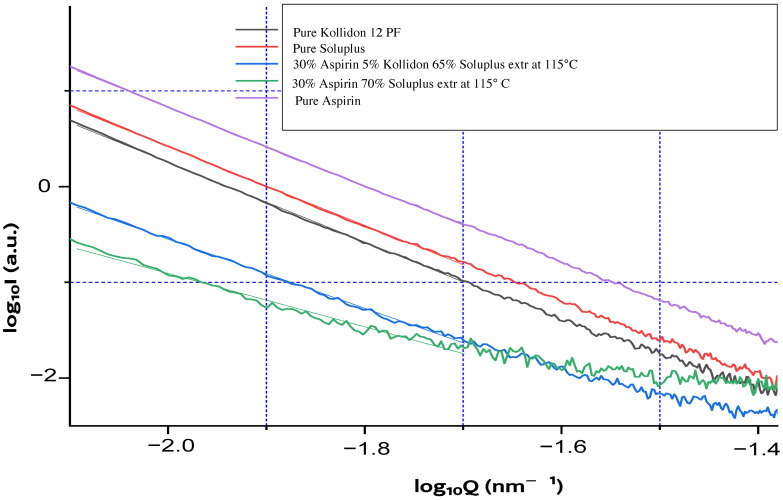
Double logarithmic SAXS plot (log(I) vs. log(q)) for pure Aspirin, Soluplus, Kollidon 12 PF, and two extruded formulations (30% ASA with Soluplus or Soluplus/Kollidon), processed at 115 °C.

**Figure 3 pharmaceutics-17-00939-f003:**
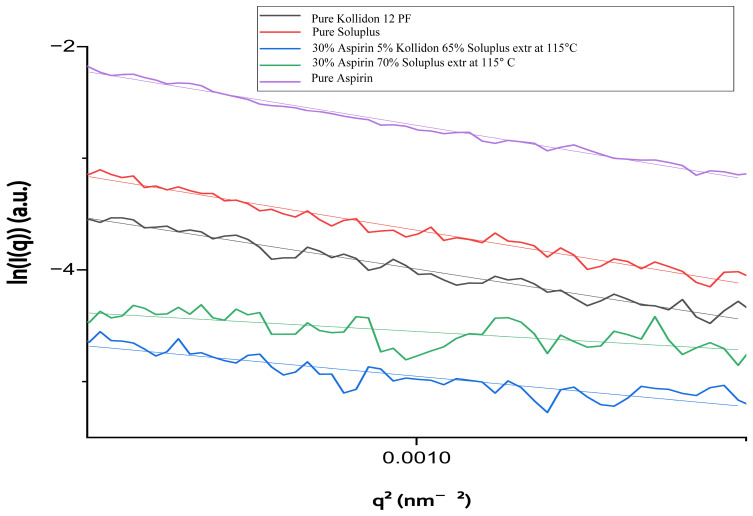
Guinier plots (ln(i) vs. q^2^) for pure components (ASA, Soluplus, Kollidon 12 PF) and extruded formulations processed at 115 °C.

**Table 1 pharmaceutics-17-00939-t001:** Radius of gyration (Rg) of raw and extruded samples.

Sample	Rg (Å) *
Kollidon^®^ 12 PF	58
Soluplus^®^	59
Acetylsalicylic acid (ASA)	62
30% ASA/70% Soluplus^®^	30
30% ASA/65% Soluplus^®^/5% Kollidon^®^ 12 PF	41

* Note: Rg = radius of gyration; values calculated from Guinier region analysis of SAXS measurements.

## Data Availability

The data supporting the findings of this study are available from the corresponding author upon reasonable request.

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
