# Peer review of "Vertical Hot-Melt Extrusion: The Next Challenge in Innovation"

_pharmaceutics, 2025, doi:10.3390/pharmaceutics17070939_

Round 1

Reviewer 1 Report

Comments and Suggestions for Authors

Please provide the full form of 'API' in the abstract.

The novelty of the paper should be clearly outlined in the introduction to ensure readers understand its unique contributions and significance within the field.

The paragraphs in the introduction are not well-connected. To enhance the flow and structure of the introduction, it should be revised and improved for better coherence.

A paragraph or text should be added to the introduction to clearly explain what is being done in the paper, providing readers with a concise overview of the objectives and scope of the work.

Please give more details about how to come this solution “So we decided to change the preferred process flow direction from the horizontal position to top-down in the vertical axis. “

Section 2 should be reorganized into two distinct subsections: 'Method' and 'Results'.

The quality of the images should be increased.

The paper currently resembles a technical report. To enhance its quality, it should include more detailed explanations and elaboration to provide greater depth and clarity.

Author Response

Dear Reviewer,

We would like to sincerely thank you for the time and effort you dedicated to reviewing our manuscript, as well as for your constructive and insightful comments. Your feedback has been extremely helpful in improving the quality and clarity of our work.

Following your suggestions and the Editor’s guidance, we have thoroughly revised the manuscript, which is now submitted as a research article rather than a review. We hope that the changes made address your concerns and meet the expectations of the journal.

Thank you again for your valuable input.

Best regards,

Maël

Reviewer 2 Report

Comments and Suggestions for Authors

The manuscript discusses the development and advantages of Vertical Hot-Melt Extrusion (HME) as an approach to improve drug solubility and bioavailability of poorly water-soluble APIs. The technology and development of the method are interesting however there are major issues that need to be addressed before this manuscript can be suitable for publication;  

The manuscript starts as a review; however it turns into presenting results without having any materials and methods; I believe this can be written as a short communication with a review in the beginning and then presenting the addition to the company’s technology but in the current form is not a research paper or a review.  

In LIne 224; if you are analysing the data using the Porod and Guineer methods you need to include more detail of what the method is; how was the data acquired etc. as well as more detail about what is presented in the graph and table.

Figure legends are incomplete: there is not enough explanation at all in Figure 3. legend.  

Does Both materials in line 46 refer to the previous paragraph? This paragraph needs rewriting as it is not understandable.  

line 55 the paragraph does not make sense. 

line 223? Only a name and reference? Does this need to be part of the materials and methods? 

Line 158   Not all API’s are sensitive to heat; are they?  

 line 97 This looks more like a list of important changes claimed by the paper while the list doesn’t provide much proof for some of these achievements; for example “simplification of extruder easy clean system” line 108; or line 109; I know the authors are aware of their work but as a reader I don’t necessarily know  what is an easy clean system and how can it be simplified! 

The figures need to be in better quality; they are grainy and seem like directly from the instrument that the experiment was performed on.  

Comments on the Quality of English Language

The introduction which must be the segway seems a little broken in places; I don’t feel there is a flow in the information.  I suggest an overview of the paper and improving the overall English. some obvious issues;

 91: the paragraph starts with a But. 

Line 93: 1 Mio$ does this mean $1 Million 

Line 138: needs rewritting:  

CO2 --> 2 needs to be subscript 

(there are many other cases of subscript that need to be fixed too.)..  

Author Response

(The authors gave the same response as above.)

Round 2

Reviewer 1 Report

Comments and Suggestions for Authors

All adjustments have made.

Author Response

Dear Reviewer, 

Thank you very much for the last review of the paper. 

The editor asked us some improvements before publication : the modifications is highlighted in yellow :

Although the effort to improve the manuscript is acknowledged, a few points still need to be addressed for the manuscript to be publishable. Please make the text less personal, avoiding the use of ‘we’ throughout the text Section 2. Rationale… is in reality part of the introduction. Please combine Discussion section presents results instead of discussing them. It is suggested that Results & Discussion are combined in one section to avoid repetitions and that results are discussed/compared in view of previous publications. The quality of figures must be improved. Please format Table according to Pharmeceutics’ rules. A thorough revision of the English language by a native speaker is also suggested to make the text more readable. 

Thank you again for your time.

Best regards, 

Maël 
